# Downscaling and validating SMAP soil moisture using a machine learning algorithm over the Awash River basin, Ethiopia

**Shimelis Sishah**[1]*, **Temesgen Abrahem**[1], **Getasew Azene**[2], **Amare Dessalew**[3], **Hurgesa Hundera**[1]

**1** Department of Geography and Environmental Studies, Arsi University, Arsi, Ethiopia, **2** Department of Geography and Environmental Studies, Debre Markos University, Debre Markos, Ethiopia, **3** Department of Geography and Environmental Studies, Bahir Dar University, Bahir Dar, Ethiopia

* shimelisgis2015@gmail.com

**Data Availability Statement:** All relevant data are within the paper and its Supporting Information files.

## Abstract

Microwave remote sensing instrument like Soil Moisture Active Passive ranging from 1 cm to 1 m has provided spatial soil moisture information over the entire globe. However, Soil Moisture Active Passive satellite soil moisture products have a coarse spatial resolution (36km x 36km), limiting its application at the basin scale. This research, subsequently plans to; (1) Evaluate the capability of SAR for the retrieval of surface roughness variables in the Awash River basin; (2) Measure the performance of Random Forest (RF) regression model to downscale SMAP satellite soil moisture over the Awash River basin; (3) validate downscaled soil moisture data with In-situ measurements in the river basin. Random Forest (RF) based downscaling approach was applied to downscale satellite-based soil moisture product (36km x 36km) to fine resolution (1km x 1km). Fine spatial resolution (1km) soil moisture data for the Awash River basin was generated. The downscaled soil moisture product also has a strong spatial correlation with the original one, allowing it to deliver more soil moisture information than the original one. In-situ soil moisture and downscaled soil moisture had a 0.69 Pearson correlation value, compared to a 0.53 correlation between the original and In-situ soil moisture. In-situ soil moisture measurements were obtained from the Middle and Upper Awash sub-basins for validation purposes. In the case of Upper Awash, downscaled soil moisture shows a variation of 0.07 $cm^3$/$cm^3$, -0.036 $cm^3$/$cm^3$, and 0.112 $cm^3$/$cm^3$ with Root Mean Square Error, Bias error, and Unbiased Root Mean Square Error respectively. Following that, the accuracy of downscaled soil moisture against the Middle Awash Sub-basin reveals a variance of 0.1320 $cm^3$/$cm^3$, -0.033 $cm^3$/$cm^3$, and 0.148 $cm^3$/$cm^3$ with Root Mean Square Error, Bias error, and Unbiased Root Mean Square Error respectively. Future studies should take into account the temporal domain of Soil Moisture Active Passive satellite soil moisture product downscaling over the study region.

**Funding:** The authors received no specific funding for this research work.

**Competing interests:** The authors have declared that no competing interests exist

## 1. Introduction

The recent advancement of microwave remote sensing opens up the possibility of retrieving soil moisture. Active microwave remote sensing has tremendous potential for detecting soil moisture due to the penetrating capabilities of radar signals directly into the surface. The upcoming Synthetic Aperture Radar (SAR) sensor with the high spatial and temporal resolution is a well-known active microwave sensor for detecting soil moisture [1]. Several studies have been conducted to retrieve soil moisture content from SAR sensors of C-band [2], X-band [3], and L-bands [4].

Active, passive, and active-passive sensors are used in microwave remote sensing in the 1 cm to 1 m range [5]. While passive microwave sensors are known as microwave radiometers, active microwave remote sensers include RADAR, LIDAR, and SONAR. Active-passive microwave sensors both have a high potential for estimating surface soil moisture content [6]. Several active-passive microwave earth-observing satellites, such as SMOS, SMAP, ASCAT, and AMSR-E, have emerged in recent years to detect soil moisture products on a global scale [7–9]. These satellite missions provide high temporal resolution soil moisture products in a matter of days, typically within 2–3 days. However, the spatial resolutions of these satellites range in tens of kilometers, making them unsuitable for local hydrological monitoring [10]. SMAP L-band radiometer satellite mission detects volumetric soil moisture of the land surface on the daily basis over a global scale. Particularly, the SMAP level 3 product provides daily volumetric soil moisture content at a depth of 5cm with a spatial resolution of 36km [7]. Although SMAP level 3 products have potential application in the hydrological process at a regional level, the coarse spatial resolution of SMAP volumetric soil moisture product limits its application at the basin scale. To use satellite-based soil moisture products of SMAP at the basin scale, the product first needs to be downscaled at the desired resolution scale.

Several downscaling algorithms namely active-passive fusion, geoinformation, and model-based have been applied to downscale coarse-resolution satellite soil moisture information to fine resolution [11, 12]. In recent years, machine learning downscaling approaches has been widely used to downscale coarse-resolution satellite soil moisture data to fine resolution [13]. According to studies, the relationship between land surface factors and the coarse-scale soil moisture product is non-linear. The usefulness of these downscaling methods was, however, limited by the complex non-linear relationship between land surface factors and coarse-scale soil moisture products [14]. Random Forest (RF) machine learning algorithm proposed by [15] and later modified by [16] has insights in describing the non-linear relationship between soil moisture and other surface physical variables. Therefore, Random Forest (RF) Machine learning approach was selected for this research work to downscale SMAP (36km x 36km) soil moisture product over the Awash River basin. However, the Random Forest model used in previous studies [14] only integrated Sentinel-2 SAR backscattering coefficient with MODIS parameters. Although understanding soil surface roughness and soil moisture variability in soil moisture estimation have great importance [17], these parameters are still neglected in the RF model to downscale SMAP products. The scientific knowledge gap identified here is the Random Forest (RF) model constraint which neglects soil surface roughness parameters in downscaling satellite-based soil moisture products. Therefore, the novelty of this research work was to improve Random Forest (RF) regression capability by integrating SAR surface roughness parameters; which were not tested in the Random Forest model yet.

This study, therefore aims to; (1) Evaluate the capability of SAR for the retrieval of surface roughness variables in the Awash River basin; (2) Measure the performance of Random Forest (RF) regression model to downscale SMAP satellite soil moisture over the Awash River basin; (3) validate downscaled soil moisture data with In-situ measurements in the river basin.

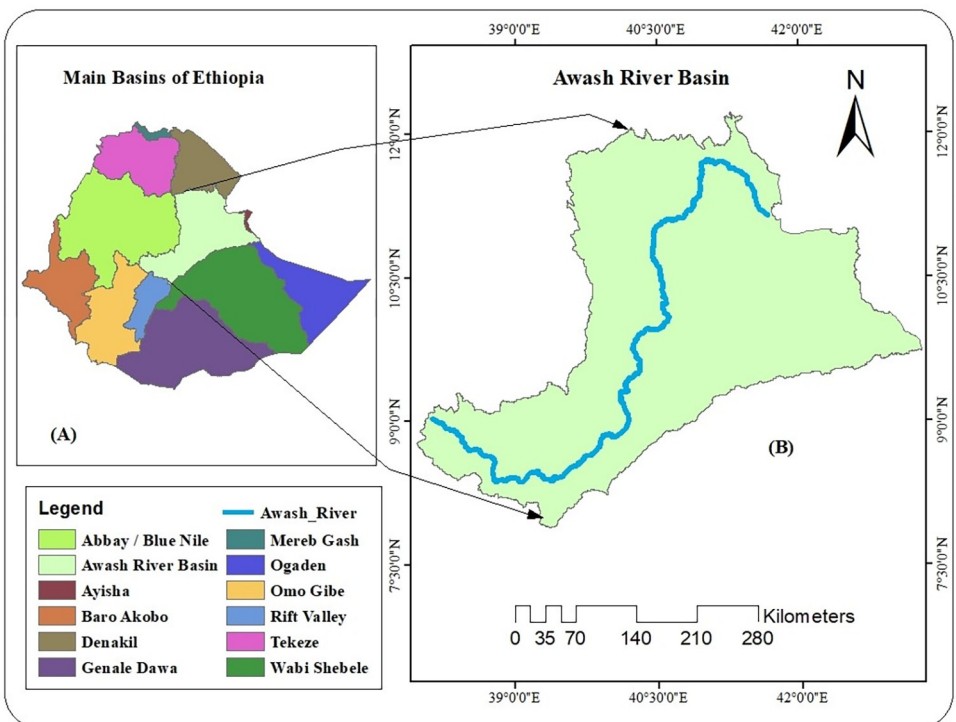

**Fig 1.** Geographic location: (A) Main Ethiopian basins, (B) Awash River Basin.

## 2. Material and methods

### 2.1. Study area description

Awash River Basin is one of the most utilized basins on the basis of the Awash river relative to the rest of the twelve main basins of Ethiopia [18]. The Awash River basin is located between 7˚ and 10˚ N latitudes and 38˚ and 41˚ E longitudes (Fig 1) The elevation of the river basin ranges from 250m in the Afar valley to 3000m in the Addis Ababa highlands above mean sea level [19]. According to [18], the Awash River basin is bounded to the west by the Blue Nile, to the southeast by the Rift Valley lakes, and the south by Wabeshebele.

### 2.2. Datasets

**2.2.1. In-situ measurements.** A random sampling method was used for the distribution of soil samples in the Awash River basin. Middle and Upper Awash sub-basins were selected for the validation of downscaled soil moisture of SMAP products. Twenty soil samples were collected from the two sub-basins in the dry season and wet season. In the Upper Awash sub-basin, validation sites are designed and placed at Melkasa Agricultural Research Institute near Adama for ease collection of field data. In doing so, ten soil samples were collected in the dry season of April 01, 2021. In addition, ten soil samples were also collected from Middle Awash sub-basin in the wet season of July 01, 2021. In the case of the Middle Awash sub-basin, validation sites are designed and placed at Amibara Agricultural Research Institute of Afar Region. Therefore, a total of twenty soil samples were collected from the two representative watersheds (Fig 2).

**2.2.2. Soil sample collection.** The characteristics of temporal and spatial variation of volumetric soil moisture content significantly affect the validation of satellite-based soil moisture

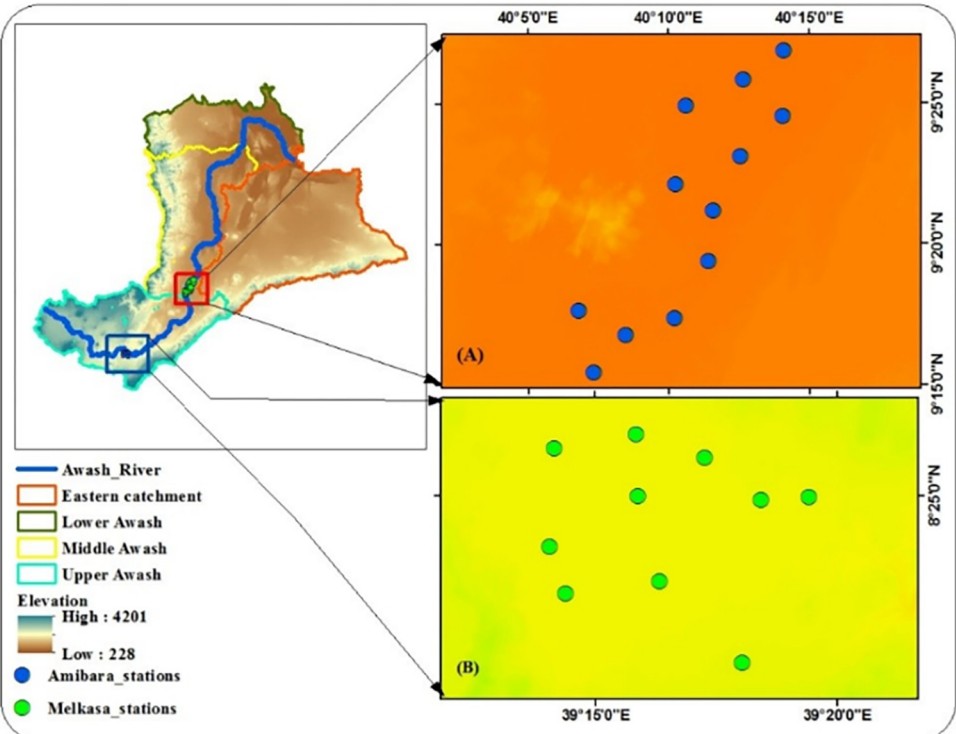

**Fig 2.** Location Map of Validation sites; (A) Amibara Stations at Middle Awash sub-basin, (B) Melkasa stations at Upper Awash sub-basin.

products [20]. Ground-based measurements of soil moisture content represent point-scale soil moisture while satellite-based soil moisture estimation quantifies the average value of the spatial grid. For this research work, twenty ground measurements were collected from the field following SMAP satellite overpass time for effective validation. Accordingly, all ground data collection was performed in line with the descending/ascending orbit path of the SMAP satellite.

Since soil varies in time and space, even over a very short distance, it is not sufficient to take samples at a single location. Therefore, a composite sampling technique was employed which involves a mixture of sub-samples at a different location. Composite sampling is to means that, collecting different soil sub-samples near the target location rather than taking a single soil sample at the desired location [21]. Then, mixing collected soil sub-samples until all the sampled soil retains a homogenous mixture which is going to be representative of the target location. Soil probe, trowel, marker, notebook, sample bag, and GPS are field materials that were used in field sample collection.

The following seven points guided the field data collection in the study area which is adapted from [21]

1. Determine appropriate locations of sample

2. Travel to appropriate sampling location

3. At the sample location, remove any plant/crop residue

4. Insert the prove to the soil with an appropriate depth of 5cm

5. Move to the next sampling site and repeat steps 3 and 4

6. Mixing different sub-sample soil with a trowel

7. Place one or two cups of mixed soil sample into the sample bag

The gravimetric method was applied to measure the amount of water content in each soil sample. In this method, the amount of water content in soil can be determined as the difference between the mass of moist soil (Fresh weight) and dry weight after the soil dried at 105-degree Celsius in oven-dry [22].

**2.2.3. Satellite data.** *Sentinel-1A SAR*. Sentinel-1 mission was first launched by the European Copernicus Program in April 2014. Its operational nature is dedicated to large-scale mapping capability and frequently revisit time. Sentinel-1 carries C-band SAR data with a frequency of 5.45 GHz. The advantage of Sentinel-1 SAR data is that its overpass period is minimal with SMAP data and this is the reason that Sentinel-1 SAR was selected in this research work. For this study, Sentinel-1 SAR images with both VV and VH polarization were acquired freely from (https://scihub.copernicus.eu/).

*SMAP*. In 2015, the National Aeronautics and Space Administration (NASA) launched the Soil Moisture Active Passive satellite (SMAP) which provides three levels (L2, L3, and L4) of soil moisture products [7]. SMAP satellites revolve around the entire world within 2–3 days and collect volumetric soil moisture with 3km, 9km, and 36km spatial resolutions [23]. For this study, level three (L3) daily soil moisture product with 36km spatial resolution was downloaded from NASA's Earth Observing System Data and Information System (https://earthdata. nasa.gov/eosdis). The main reason that 36km daily soil moisture product preference for this research work was its coarse spatial resolution relative to others. On the other hand, the SMAP satellite has minimal overpass time over the Sentinel-1 SAR and MODIS satellites.

*MODIS*. MODIS satellites have two sensors, Terra and Aqua operated on solar synchronous orbit. Terra sensors revolve around the world at 10:30 am and 10:30 pm descending and ascending local times respectively. While Aqua sensor revisits time was at 1:30 am and 1:30 pm descending and ascending local times respectively. A combination of Terra and Aqua sensors provides MODIS data of land surface variables including LST, EVI, NDVI, LAI, etc. with a different overpass period. MYD11A1, MOD13A2, and MCD15A3H data were downloaded from NASA's Earth Observing System Data and Information System (EOSDIS) to be used in this study.

*Soil Grids*. International soil reference and information center (ISRIC) provided 1km spatial resolution gridded soil data over the entire globe [24]. Soil texture data were used as predictor variables by [25] to downscale both active and passive microwave soil moisture products. Therefore, based on these insights five soil-gridded datasets such as clay fraction, sand fraction, silt fraction, bulk density, and soil organic carbon were selected as predictor variables and downloaded from (http://data.isric.org/) website freely.

*Precipitation*. Climate Hazard Group Infrared Precipitation (CHIRPS) data is a long-term temporal coverage data (1981 to present). It gives precipitation data on a worldwide scale with 0.05 x 0.05 degrees, which is about five kilometers (5km) [26]. CHIRPS data archive provides rainfall data on a daily, monthly, and annual basis. For this research work, the mean monthly rainfall data of the year 2021 was downloaded from the (https://earlywarning.usgs.gov/fews) site freely. Table 1 depicts satellite data type, source and resolution used in this research work.

## 2.3. Methods

**2.3.1. Methodological design.**   The methodological approaches of this research work can be seen as a five-stage process. At the first stage, land surface variables such as LST, LAI, NDVI, EVI, Precipitation, surface roughness (surface height and effective correlation length),

**Table 1. Characteristics of satellite data collected over the study region.**

| Satellite | Product ID | Spatial resolution | Temporal resolution | Source |
|---|---|---|---|---|
| SMAP | SPL3SMP | 36km | Daily | https://earthdata.nasa.gov/eosdis |
| MODIS | MOD11A1 | 1km | Daily | https://earthdata.nasa.gov/eosdis |
| | MCD15A3H | 1km | 8-day | https://earthdata.nasa.gov/eosdis |
| | MOD13A2 | 1km | 16-day | https://earthdata.nasa.gov/eosdis |
| | MOD13A2 | 1km | 16-day | https://earthdata.nasa.gov/eosdis |
| | MOD16A2 | 1km | 8-day | https://earthdata.nasa.gov/eosdis |
| Soil Grids | Soil grids | 1km | - | http://data.isric.org/ |
| Sentinel_1A | SLC SM | 10m | 5-day | https://scihub.copernicus.eu/ |
| MODIS | ET | 1km | Monthly | https://earlywarning.usgs.gov/fews/ |
| CHIPRS | CHIRPS 2.0 | 5km | Daily | https://earlywarning.usgs.gov/fews/ |

soil texture, bulk density, and soil organic carbon were retrieved from MODIS, Soil grids, CHIRPS, and SAR products. At the second stage, Random Forest regression was constructed between surface soil moisture of SMAP and land surface variables derived from MODIS, CHIRPS, Soil Grids, and SAR products. At the third stage, model diagnostic was conducted to indicate the model importance of each of the land surface variables. At the fourth stage, the RF-based downscaling method was applied to the best fit relationship model of surface variables. Finally, validation of downscaled soil moisture of SMAP with in-situ measurement has been applied. The flow chart in (Fig 3) depicts the general steps used in this research work that is further described in the next section.

**2.3.2. Soil surface roughness estimation.** Determining and incorporating the effects of soil surface roughness is very important for monitoring soil moisture content [27]. Soil surface roughness is expressed in terms of effective correlation length (L) and standard deviations of

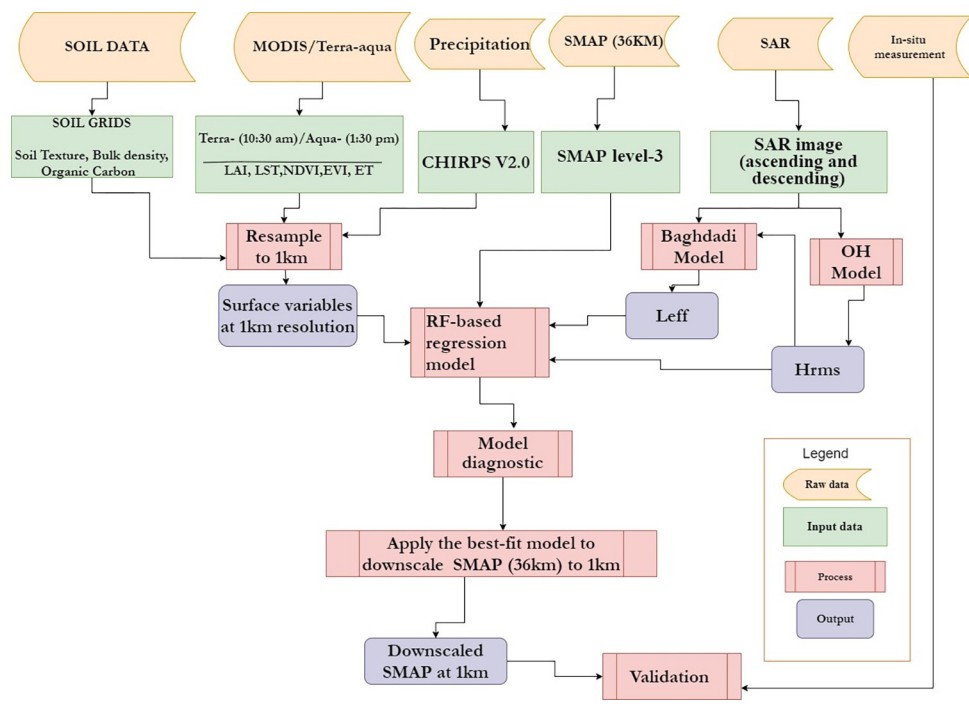

**Fig 3. Methodological flow chart.**

surface height (h) [28]. A semi-empirical polarized backscattering equation inverted by [29] was used in this research work to derive the standard deviation of surface height (h). [27] applied the inversion of this model for the estimation of soil surface height (h). Here is the [28] model and its inversion for surface height (h) measurement in Eq 2.

$$q = \frac{\sigma \, soil_{VH}}{\sigma \, soil_{VV}} = 0.95(0.13 + \sin 1.5\theta)^{1.4}(1 - \epsilon^{-1.3(K.h_{rms})^{0.9}}) \tag{1}$$

$$h_{rms} = \frac{\left\{ \frac{-1}{1.3} \ln\left[ 1 - \frac{q}{0.95(0.13 + sin1.5\theta)^{1.4}} \right] \right\}^{1.111}}{k} \tag{2}$$

Where $q$ is cross-polarized ratios, $\theta$ is incidence angle in degree, $K$ is the wavelength of SAR band, $h_{rms}$ is the surface height at root mean square error. The surface correlation length (L) parameter was estimated based on the assumption that correlation length and surface height at root mean square has a linear relationship over agricultural fields [30]. The relationship between these surface parameters was proposed by [31] which describes effective correlation length as a function of surface height at root mean square. For this research work, a model developed by [32] was used for surface length estimation. Here is the equation for surface correlation length (Eq 3).

$$L_{eff} = (\alpha * h_{rms})^{\beta} \tag{3}$$

Where, $L_{eff}$ is effective correlation length, $h_{rms}$ is the surface height at root mean square, $\alpha$, and $\beta$ are coefficients which can be calculated from incidence angle ($\theta$) and backscattering coefficient of VV- polarization ($\sigma_{soil\_VV}$) as follows.

$$\alpha = \delta(sin\theta)^{\mu} \tag{4}$$

$$\beta = \eta\theta + \xi \tag{5}$$

Although $\delta$ and $\xi$ parameters are derived from calibration coefficient of polarization, η and $\mu$ parameters are constant and independent from polarization. In this case, the values of $\delta$, $\xi$, η and $\mu$ parameters are as follows:

$$\delta_{soil_{vv}} = 3.829, \quad \xi_{VV} = 1.222, \quad \eta_{VV} = -0.0025, \quad \mu_{VV} = -1.744$$

**2.3.3. Random Forest (RF) regression.** Random Forest (RF) algorithm is an ensemble machine learning approach proposed by [15] and later modified by [16]. Based on decision trees, it can be used for both classification and regression of different variables. To improve the predictability of response variables in individual decision trees, the data are split into homogenous units of nodes. When bootstrapped data is predefined, the Random Forest (RF) algorithm integrates separate decision trees. Consequentially, the predicted values of continuous variables are the mean values of all fitted variables as a result of each bootstrapped sample [16]. [33] have shown that the Random Forest (RF) algorithm can be used to establish a soil moisture relationship with surface variables, which is why the Random Forest (RF) approach was used in this research. Eq 6 shows that the Random Forest (RF) model is a dependent variable that is directly influenced by independent variables.

$$h(X, \theta t), t = 1, 2, 3 - - - tn \tag{6}$$

where: θt -Independent variable generated from regression trees

X-Independent variable

t-Number of regression trees

**2.3.4. Downscaling SMAP soil moisture as a function of land surface variables.** The non-linear function was constructed between land surface variables and coarse resolution soil moisture data of SMAP using the Random Forest regression model. The built relationship model is spatial scale independent and the non-linear model was directly applied to the land surface variable resolution to predict fine resolution soil moisture data. Therefore, the proposed downscaling function developed using RF-regression for satellite-based soil moisture product of SMAP was described in Eq 7.

$$SM_{1km} = F_R(NDVI_{1km}, LAI_{1km}, LST_{1km}, EVI_{1km}, Hrms_{1km}, Leff_{1km}, bld_{1km}, cly_{1km}, ET_{1km}, slt_{1km}, oc_{1km}, prec_{1km})(7)$$

Where, **SM** is soil moisture at fine resolution, **$F_R$** is Random Forest regression function, and *NDVI, LAI, LST, EVI, Hrms, Leff, bld, cly, ET, slt, oc, and prec* are normalized difference vegetation index, leaf area index, enhanced vegetation index, surface height at root mean square, effective correlation length, bulk density of the soil, clay fraction, evapotranspiration, silt fraction, organic carbon, and precipitation at 1km spatial resolution respectively.

**2.3.5. Validation with the in-situ soil moisture measurement.** Downscaled soil moisture product has been validated with in situ soil moisture measurements. The effectiveness of the downscaled soil moisture product can be quantified with Root Mean Square Error (RMSE), Unbiased Root Mean Square Error (ubRMSE), Bias error, and Pearson Correlation coefficient (R). RMSE and ubRMSE which range between zero and positive infinity ($0\sim +\infty$) show the variance and unbiased variance between satellite soil moisture and in situ measurements respectively. Bias error ($-\infty \sim +\infty$) is a measure of soil moisture difference between satellite and in situ measurements. Meanwhile, the relationship between satellite and in situ soil moisture can be measured with a Pearson correlation coefficient between ($-1\sim1$). The following equations (Eqs 8–11) show, RMSE, ubRMSE, Bias error, and Pearson correlation coefficient (R) respectively.

$$RMSE = \sqrt{\frac{\sum_{i=1}^{N}(\theta_i - \theta_s)^2}{N}} \tag{8}$$

$$ubRMSE = \sqrt{\frac{\sum_{i=1}^{N}((\theta_i - Bias) - \theta_s)^2}{N}} \tag{9}$$

$$Bias\ error = \frac{\sum_{i=}^{N}(\theta_i -)}{N} \tag{10}$$

$$R = \frac{N\sum(\theta_i\theta_s) - \sum\theta_i\sum\theta_s}{[-(\sum(][N(\sum(\theta_i^2 - (\theta_s))]} \tag{11}$$

Where, **$\theta_s$** is the soil moisture content of SMAP, **$\theta_i$** is In-situ measurement of soil moisture, **N** is the number of the sample point.

## 3. Results

### 3.1. Soil surface roughness estimation

Soil surface roughness of the study region was generated using the Oh model for April 01, 2021. The results of surface height for the Awash River basin range from a minimum of 0.13

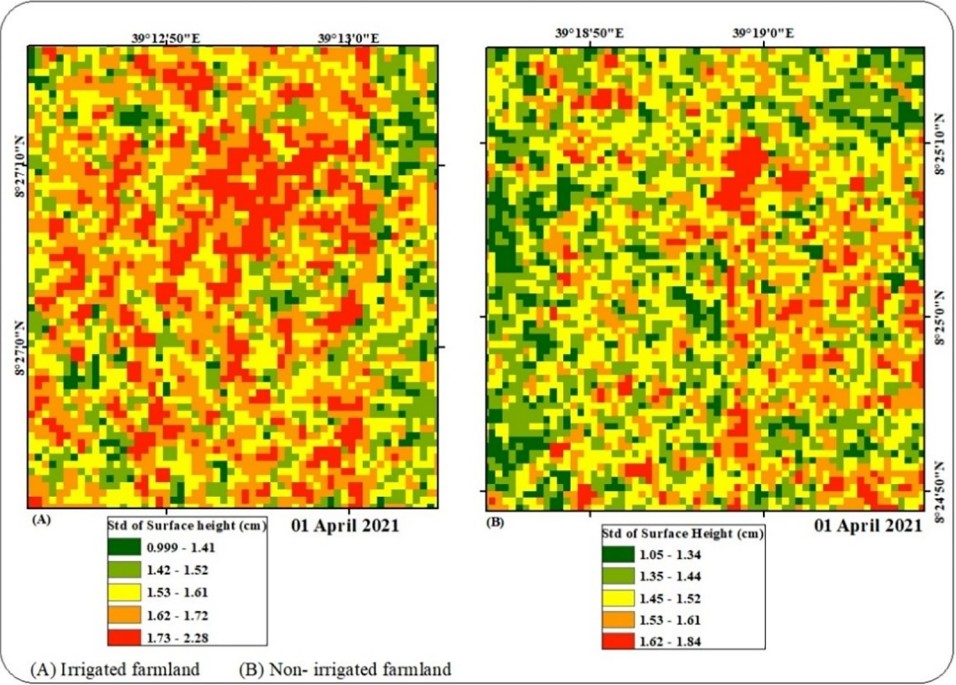

**Fig 4. Surface height.**

cm to a maximum of 2.993 cm. For ease of clarification of surface height, irrigated and non-irrigated farmland sample plots were selected from the Awash River basin. As presented in (Fig 4A) irrigation has the higher surface height ($h_{rms}$) that ranges from a minimum of 0.99 cm to a maximum of 2.28 cm. Whereas non-irrigated farmland experiences smooth roughness compared to irrigated farmland with a surface height that ranges from 1.05 cm to 1.84 cm (Fig 4B). The main reason that soil surface roughness exhibited increment at irrigated farmland was that the soil has been under influence of agricultural activities such as plowing. On the other side, non-irrigated farmland was free from agricultural activities, hence the data collection was performed during the off-season of April 1, 2021.

The results of the correlation length of the Awash River basin range from 0.042 cm to 6.63 cm. As of the standard deviation of surface height, the correlation length of the soil surface in the river basin was illustrated from the sample plots of irrigated and non-irrigated farmland. As indicated in (Fig 5A), a higher number of effective correlation lengths was observed at irrigated farmland over non-irrigated farmland sample plots. Effective correlation ($l_{eff}$) length of irrigated farmland was range from 0.74 cm to 3.87 cm. Sample plot of non-irrigated farmland (Fig 5B) exhibited minimum effective correlation ($l_{eff}$) relative to irrigated farmland sample plots that range from 0.825 cm to 2.52 cm.

## 3.2. Out of Bag error (OOB) in Random Forest regression

In Random Forest regression, each of the prediction trees is constructed from the original data using Bootstrap or the subsampling technique. Observations obtained from model prediction variables such as NDVI, LST, EVI, LAI, Precipitation, Bulk density, surface length, surface height, Silt fraction, Soil organic carbon, Clay fraction, and Evapotranspiration are boot-strapped or subsampled randomly. Observations that are not included in bootstrap are said to be out-of-bag observation which is very important in computing out-of-bag error. The out-of-

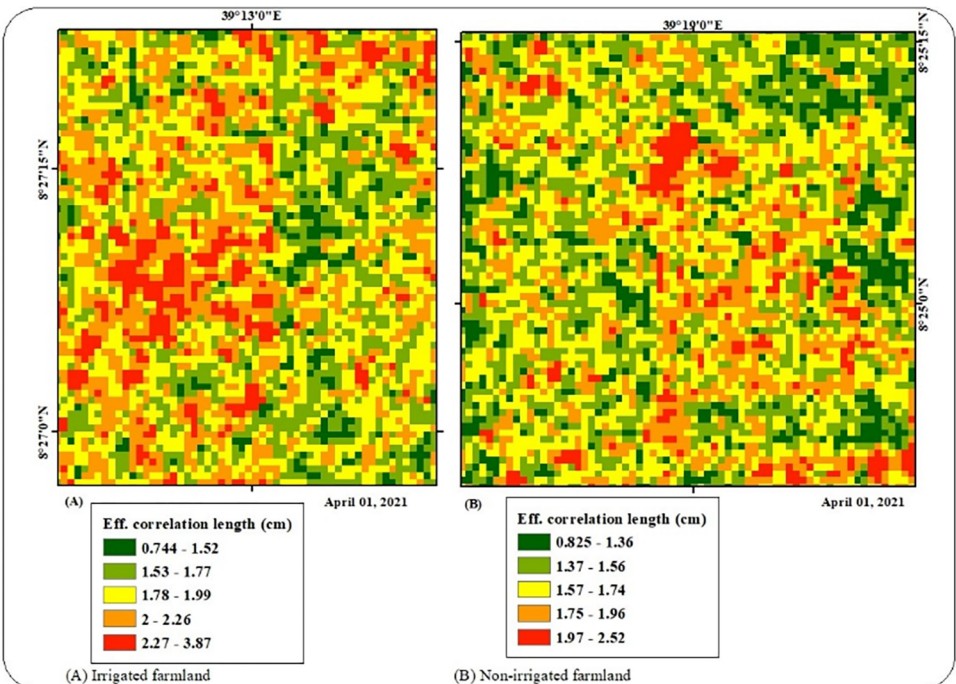

**Fig 5. Effective correlation length.**

bag error is the error that denotes the prediction performance of the Random Forest regression model. According to [34], the out-of-bag error (OOB) was getting stabilized with a minimum of 250 regression trees, and 1000 trees were more than enough for the wide dataset. On contrary, [35] argues that Random Forest regression with 500 trees and 1000 trees resulted in the same out-of-bag error (OOB) error. Based on this literature the number of trees was set to 500 trees for this research work. The prediction performance of Random Forest regression can be quantified with Mean Squared Error (MSE) as a function of several regression trees. As presented in (Fig 6), the Mean Squared Error (MSE) of the graph shows that the Random Forest regression model is negligible when the number of regression trees goes to zero. As the number of regression trees increased, the Mean Squared Error (MSE) of the prediction model decreased and the Random Forest model gets its optimal out-of-bag error (OOB).

## 3.3. Variable importance in RF regression

To interpret the fitted Random Forest regression model, it is important to measure the importance of predictor variables [16, 35]. Random Forest Regression produces two important measures that explain the predictive power of individual parameters. Increased Mean Squared Error (IncMSE) and Increased Node Purity Index (IncNudePurity) are the two qualitative measures of the prediction power of model parameters. Increased Mean Squared Error (IncMSE) measures the effect of the individual parameter when it is permuted randomly. The higher the values of Increased Mean Squared Error (IncMSE) mean the most important the variable is in the Random Forest regression model. It is a measure of by how much removing the most important variable decreases the accuracy of Random Forest regression or including the most important variable increases the model prediction. Thirteen (13) predictors were used to predict the coarse resolution of SMAP (36km) soil moisture products to fine spatial resolution (1km). These predictors were; Land surface temperature (day time), Normalized

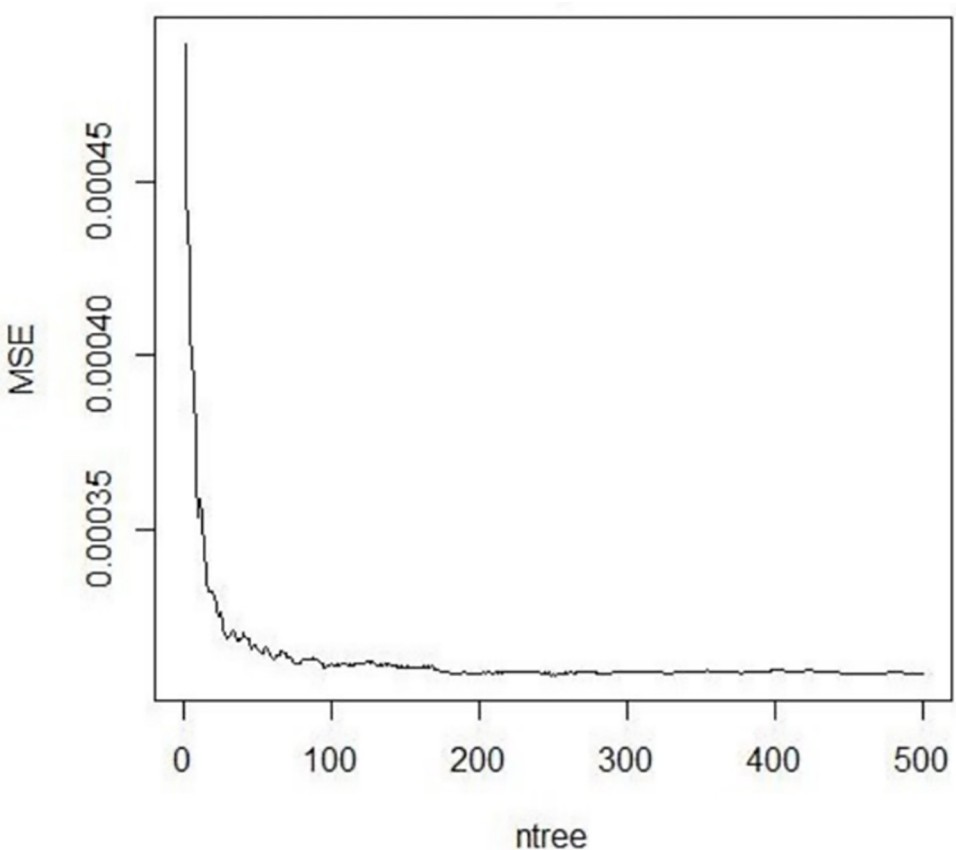

**Fig 6. Model prediction error.**

difference vegetation index, Land surface temperature (night time), Enhanced vegetation index, leaf area index, soil surface height, effective correlation length of the soil surface, Precipitation, Bulk density, Silt fraction, Soil organic carbon, Clay fraction, and Evapotranspiration. Increased Mean Squared Error (IncMSE) measurement of predictor variables shows that Land surface temperature (day time) was the most model importance variable. Whereas Evapotranspiration was the least model importance variable (Fig 7).

Besides, the Increased Node Purity Index (IncNudePurity) measures the homogeneity of splinting bootstrapped samples to the desired variables. For the node purity importance, a split with an increase of purity is said to be very important for model prediction. Increased Node Purity Index (IncNudePurity) measurement of predictors indicates that Land surface temperature (night time) was persistent at splitting nodes. In increasing order of node purity index; Leaf area index, Clay fraction, effective correlation length, soil surface height, Enhanced vegetation index, Evapotranspiration, Normalized difference vegetation index, Soil organic carbon, Silt fraction, Land surface temperature (day time), Precipitation, and Land surface temperature (night time) respectively (Fig 8).

## 3.4. Downscaling results

Land surface variables such as; Leaf area index, Clay fraction, effective correlation length, soil surface height, Enhanced vegetation index, Evapotranspiration, Normalized difference vegetation index, Soil organic carbon, Silt fraction, Land surface temperature (day time),

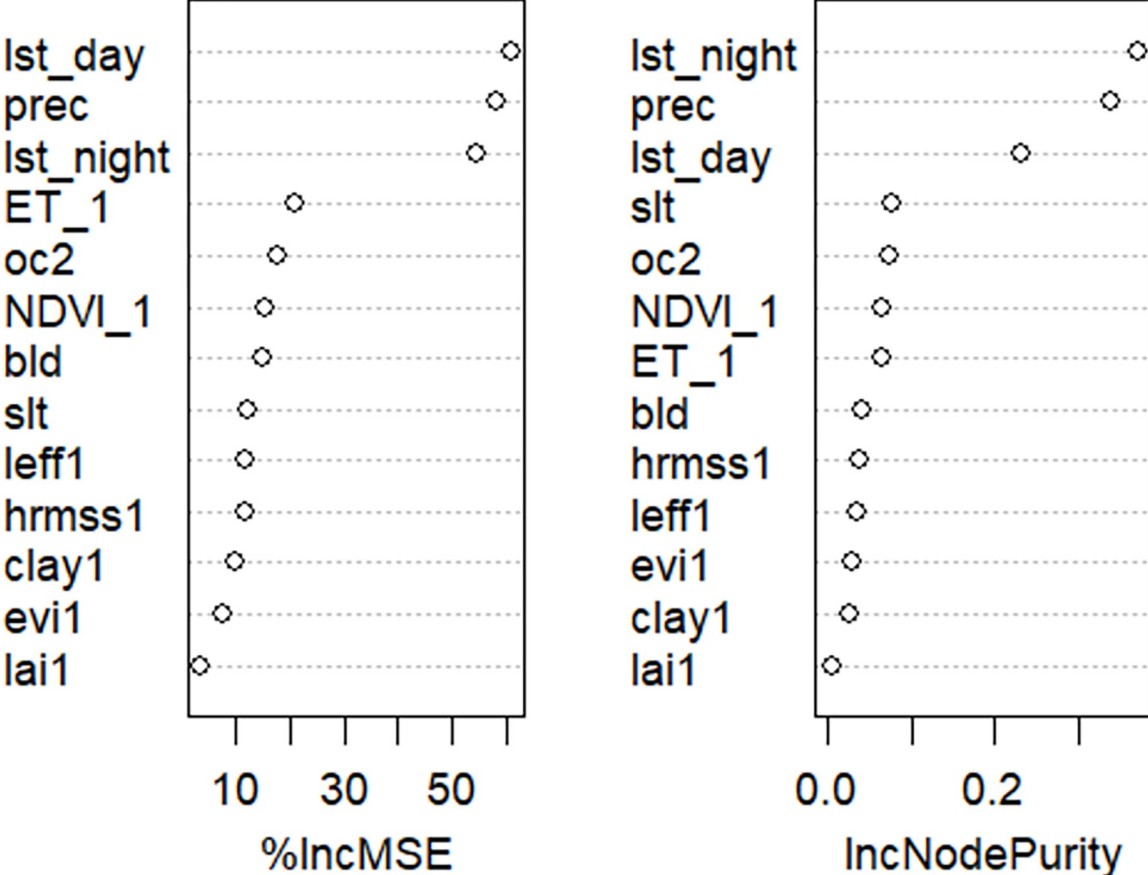

**Fig 7. Variable importance in model prediction.**

Precipitation, and Land surface temperature (night time) were used as proxy data to downscale the coarse resolution of SMAP (36km x 36km) to fine resolution (1km x 1km). The R software package of Model Map was applied to create a raster layer from predictor variables at 1km spatial resolution. As indicated in (Figs 9 and 10), the downscaled soil moisture of SMAP at a spatial resolution of 1km range from the minimum of 0.077cm$^3$/cm$^3$ to the maximum of 0.139 cm$^3$/cm$^3$. The higher volumetric soil moisture value was observed at the south and southwestern part of the river basin. The North and northwestern part of the study region was experienced with low volumetric soil moisture value. The results of downscaling reveal that the spatial heterogeneity of the SMAP soil moisture can be captured by the downscaled SMAP soil moisture. The downscaled SMAP also has a strong spatial correlation with the original SMAP, allowing it to provide more detailed soil water information than the original 36 km resolution.

[36] showed comparable results, demonstrating that downscaled soil moisture of SMAP can accurately reflect soil moisture information over coarse resolution soil moisture of SMAP

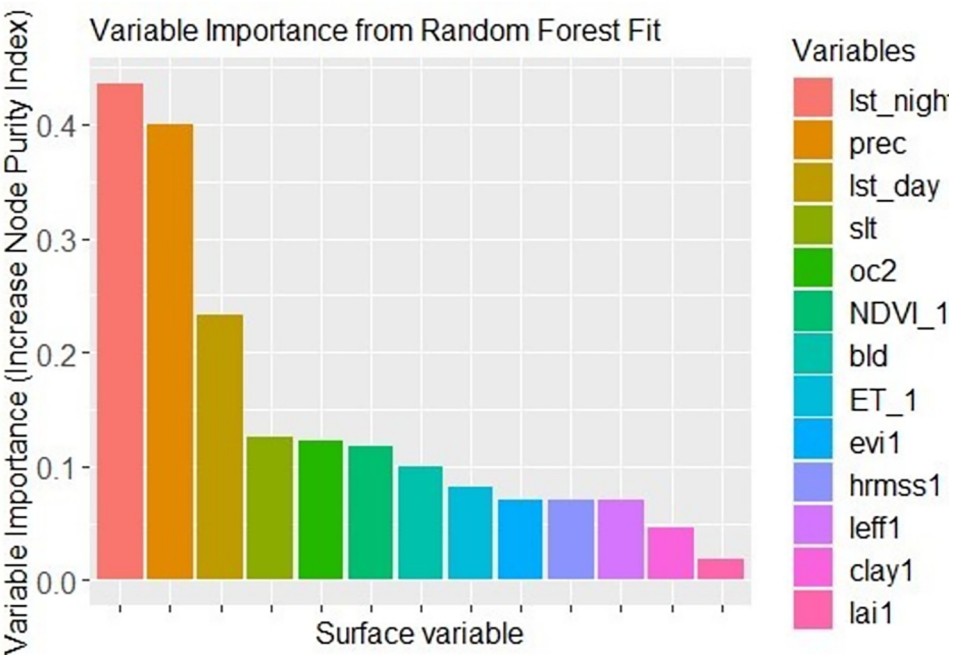

**Fig 8. Increased node purity index measurement from Random Forest fit.**

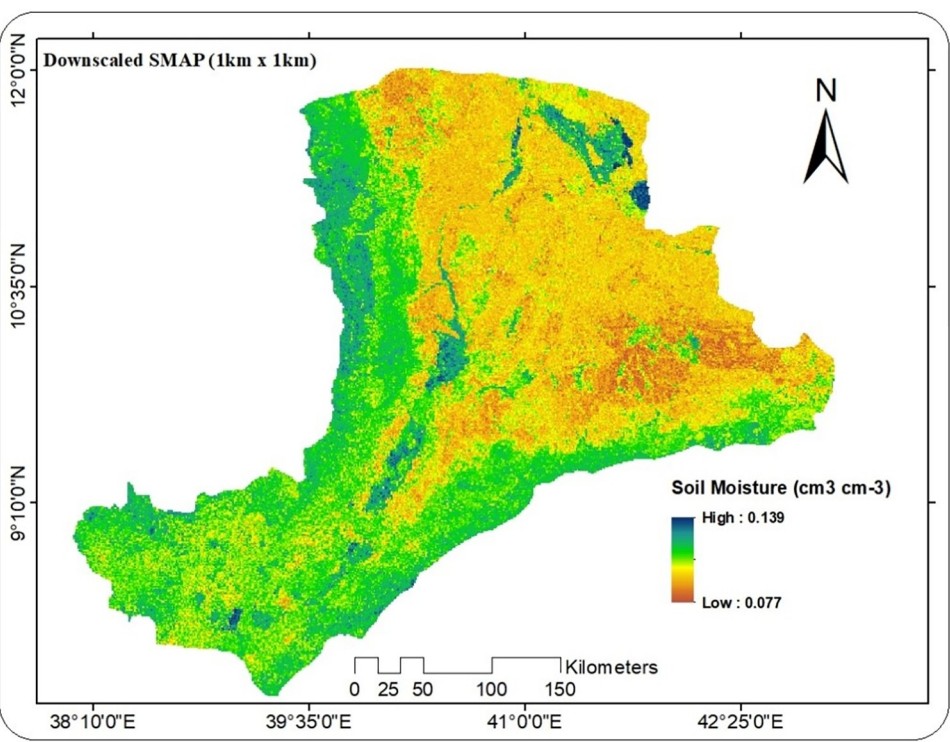

**Fig 9. Final downscaled soil moisture of SMAP (1km x1km) over Awash River basin.**

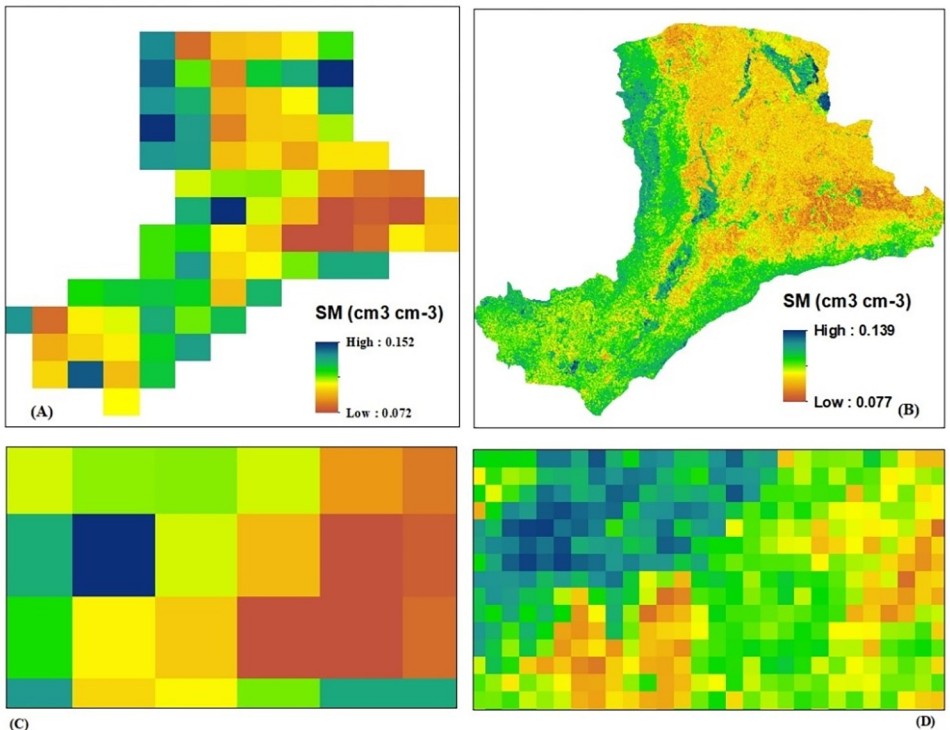

**Fig 10.** SMAP soil moisture; (A) original SMAP (36km x 36km) at basin scale, (B) downscaled SMAP (1km x 1km) at basin scale; (C) original SMAP (36km x 36km) at pixel level; (D) downscaled SMAP (1km x 1km) at pixel level.

(36km x 36km). In this work, they used NDVI, LST, and LAI as Random Forest Model predictors to estimate fine resolution soil moisture. Similarly, [37] contend that SMAP downscaled soil moisture (1km x 1km) may efficiently characterize soil moisture information when compared to in-situ data. In this regard, the findings of this study were in agreement with prior scholarly studies to the point where soil moisture estimates at 1 km resolution can provide extensive information on the spatial distribution and pattern over the regions under consideration. For practical applications and modeling in large watersheds with limited in situ data, such as the Awash River basin, higher-resolution soil moisture data are required.

## 3.5. Validation

Validation of downscaled soil moisture in the Awash River basin was conducted over two validation sites of the Upper and Middle Awash sub-basins. As a result, downscaled soil moisture of SMAP product was well agreed with In-situ soil moisture measurements from Upper and Middle awash sub-basins with Pearson correlation coefficients of 0.69 and 0.57 respectively. Downscaled soil moisture of SMAP showed a variance from In-situ measurements with an RMSE value of 0.07 $cm^3/cm^3$ over Upper Awash validation sites. The variance of downscaled soil moisture from In-situ measurements was higher over the Middle Awash validation site with an RMSE value of 0.1320 $cm^3/cm^3$. In addition, the Unbiased variance of both validation sites was computed with the statistical measure of Unbiased Root Mean Squared Error (UbRMSE). It was found out that, downscaled soil moisture of SMAP was unvaried from In-situ measurements over Upper Awash sub-basin with UbRMSE value of 0.112 $cm^3/cm^3$ on the date of April 01, 2021. Meanwhile, Unbiased Root Mean Square Error measurement over Middle Awash sub-basin was said to be larger than Upper Awash sub-basin; that is 0.148 $cm^3/cm^3$.

**Table 2. Summary of validation results for the downscaled soil moisture against site measurements.**

| Samples | Bias error (cm³/cm³) | RMSE (cm³/cm³) | UbRMSE (cm³/cm³) |
|---|---|---|---|
| Upper Awash Sub-Basin Validation results over Melkasa Station on April 01, 2021 | | | |
| sample 1 | -0.0024 | 0.0024 | 0.0048 |
| sample 2 | 0.0002 | 0.0002 | 0.0004 |
| sample 3 | -0.0107 | 0.0107 | 0.0214 |
| sample 4 | -0.0110 | 0.0110 | 0.0220 |
| sample 5 | -0.0070 | 0.0070 | 0.0140 |
| sample 6 | -0.0018 | 0.0018 | 0.0037 |
| sample 7 | 0.0002 | 0.0002 | 0.0004 |
| sample 8 | -0.0030 | 0.0030 | 0.0060 |
| sample 9 | -0.0070 | 0.0070 | 0.0140 |
| sample 10 | 0.0061 | 0.0061 | 0.0123 |
| **Total** | **-0.036** | **0.0702** | **0.112** |
| Middle Awash Sub-Basin Validation results over Amibara Station on July 01, 2021 | | | |
| sample 1 | -0.0119 | 0.0119 | 0.01853 |
| sample 2 | 0.0051 | 0.0051 | 0.01938 |
| sample 3 | 0.0037 | 0.0037 | 0.01264 |
| sample 4 | 0.0550 | 0.0550 | 0.04988 |
| sample 5 | -0.0134 | 0.0134 | 0.02342 |
| sample 6 | -0.0571 | 0.0571 | 0.05531 |
| sample 7 | -0.0084 | 0.0084 | 0.00574 |
| sample 8 | 0.0039 | 0.0039 | 0.00214 |
| sample 9 | -0.012 | 0.0127 | 0.02048 |
| sample 10 | 0.0027 | 0.0027 | 0.01275 |
| **Total** | **-0.0331** | **0.1320** | **0.1484** |

Table 2 summarizes statistical validation measures such as Bias error, RMSE, and UbRMSE between downscaled soil moisture of SMAP and In-situ soil moisture measurements over the Awash River basin.

## 4. Discussion

The ability of Sentinel-1A SAR data to estimate soil surface roughness was investigated in this study. Different techniques can be used to extract surface soil moisture from SAR data because it is sensitive to radar backscattering [38]. But in addition to soil moisture, radar backscattering is also susceptible to other time- and space-varying factors including vegetation and soil roughness [39, 40]. In this study, soil surface roughness was estimated using Sentinel-1 SAR data. Surface height and length were estimated from Sentinel-1A Synthetic Aperture Radar (SAR) with both Vertical release vertical receive (VV) and Vertical release horizontal receive (VH) polarizations. The results showed the potential of Sentinel-1 SAR data for soil surface roughness estimation. The Awash River basin's surface height thus varies from a minimum of 0.13 cm to a maximum of 2.993 cm. Additionally, it was shown that the Awash River basin's effective correlation length ranges from 0.042 cm to 6.63 cm. According to many researchers [17, 38], taking into account soil roughness factors might further increase the precision of soil moisture downscaling using Random Forest (RF) regression. The Oh and Baghdadi models were also used to calculate the research site's surface height and effective correlation length in order to increase the accuracy of the soil moisture downscaling. Surface roughness parameters (surface height at root mean square error and effective correlation length) introduced in

Random Forest regression showed good model performance. As indicated in [33, 41], the integration of MODIS land surface variables with Sentinel 1 backscattering coefficient shows good Random Forest (RF) model performance. On contrary, this research work tried to integrate MODIS land surface variables with Sentinel 1A derived surface roughness variables (surface height at root mean square error and effective correlation length). As a result, surface roughness parameters introduced in this research work showed satisfactory model performance. Therefore, the newly introduced model variables (soil surface height and effective correlation length) were increased the robustness of Random Forest (RF) regression to downscale satellite-based soil moisture of SMAP to fine spatial resolution.

The predictive power of individual land surface variables was evaluated with Increased Mean Square Error (IncMSE) and Increased node Purity Index (IncPurity). According to [42] the higher the values of both measurements reveal the relative importance of individual land surface variables. Supporting this statement, Land surface temperature (LST day time) was the most model importance variable in Random Forest (RF) regression and LAI was the least model importance variable. The main reason that LST (day time) is the most model important variable was attributed to its strong relation with SMAP soil moisture product relative to other land surface variables. It was found out that, SMAP satellite soil moisture was explained by LST (day time) with a coefficient of determination ($R^2$) value of 0.228. On the other side, the homogeneity of splinting bootstrapped samples to the desired variable was measured with the Increased Node Purity Index (IncNudePurity). It was found out that, Leaf Area Index (LAI) and Land surface temperature (LST night time) were the least and the most important variables for the model prediction respectively.

Downscaled soil moisture values at fine resolution showed a very small variation from original SMAP soil moisture values. A similar finding was reported by [43–45] that revealed validation results of SMAP soil moisture. In their study, soil moisture of SMAP showed a small variation over core In-situ soil moisture measurements. In this research work, downscaled soil moisture of SMAP product was validated against In-situ soil moisture measurements taken from two main sub-basins of the Awash River basin. Considerably, downscaled soil moisture values exhibited a variation with an RMSE value of 0.015 cm$^3$/cm$^3$ and 0.010 cm$^3$/cm$^3$ from original SMAP soil moisture values over Upper and Middle Awash sub-basins respectively. The main reason behind this soil moisture variation was that the coarse resolution of the SMAP (36km) product does not fit with the natural boundary of the Awash River basin. Areas beyond the extent of coarse resolution of SMAP (36km) were predicted from the nearest pixel values. Although estimating soil moisture values to a specified extent is vital, predicting soil moisture from existing cell values is not recommended due to spatial heterogeneity of soil moisture within a small distance.

## 5. Conclusion

In this study, satellite-based soil moisture product of SMAP (36km x 36km) was downscaled to fine spatial resolution (1km) over the Awash River basin. Random Forest algorithm developed by [44] was applied to predict soil moisture information at fine resolution in the river basin. NDVI, EVI, LST, LAI, Soil grids, Precipitation, Evapotranspiration, and soil surface roughness were used as predictor variables in the Random Forest regression model. In-situ soil moisture measurements were taken from twenty (20) validation sites to compare downscaled soil moisture of SMAP (1km) with ground soil moisture measurements. Statistical techniques such as RMSE, UbRMSE, Bias error, and Pearson correlation coefficient were used to evaluate the performance of downscaled soil moisture over proposed validation sites. The conclusion drawn from this research findings are presented as of each of the research objectives: -

1. The capability of sentinel 1A SAR data was evaluated to understand the surface roughness of the Awash River basin. Soil surface roughness can be expressed in terms of surface height ($h_{rms}$) and effective correlation ($l_{eff}$) length. Surface height and length were estimated from Sentinel-1A Synthetic Aperture Radar (SAR) with both Vertical release vertical receive (VV) and Vertical release horizontal receive (VH) polarizations. The results of surface height for the Awash River basin range from the minimum of 0.13 cm to the maximum of 2.993 cm. In addition, it was found out that, Effective correlation length of the Awash River basin range from 0.042 cm to 6.63 cm.

2. Performance of Random Forest regression model was tested to predict soil moisture information from coarse-scale satellite-based soil moisture product at a land surface variable resolution over Awash River basin. Subsequently, the Random Forest (RF) regression model has the potential to bootstrap land surface variables to reduce the overall prediction error. Land surface variables are ranked in order of model importance based on statistical measures of Increased Mean Square Error (IncMSE) and Increased Node Purity Index (IncPurity). This model also explains the association between predicted and observed soil moisture of SMAP with a Pearson correlation coefficient of 0.76. Predicted soil moisture values obtained from surface variables using Random Forest regression have deviated with an RMSD value of 0.02.

3. Statistical measures such as RMSE, UbRMSE, Bias error, and Pearson correlation coefficient were applied to evaluate the performance of downscaled soil moisture with in-situ soil moisture measurements. As a result, downscaled soil moisture of SMAP exhibit a total variation of 0.0702 cm$^3$/cm$^3$ and 0.1320 cm$^3$/cm$^3$ over In-situ soil moisture measurements taken from Upper and Middle Awash sub-basins respectively. Furthermore, downscaled soil moisture of SMAP was different over In-situ soil moisture measurements taken from Amibara stations of Middle Awash Sub-basin with a bias error of -0.033 cm$^3$/cm$^3$. Similarly, the bias error estimation over Melkasa sites of Upper Awash revealed that downscaled soil moisture from In-situ soil moisture measurements varied with a bias error of -0.036 cm$^3$/cm$^3$. In addition, it was found that unbiased variance of downscaled soil moisture of SMAP over In-situ soil moisture measurements was 0.112 cm$^3$/cm$^3$ and 0.148 cm$^3$/cm$^3$ over Upper and Middle Awash sub-basins respectively. Validation results of findings reveal that downscaled soil moisture of SMAP was strongly correlated with In-situ soil moisture measurements from all validation sites.

## Supporting information

**S1 Table.**
(CSV)

**S2 Table.**
(CSV)

**S3 Table.**
(CSV)

## Acknowledgments

The authors would like to thank the European Space Agency (ESA) and the United States geological Survey (USGS) for providing the satellite data. We are also grateful to the Awash Basin Development Office for the soil moisture measurement instruments.

## Author Contributions

**Conceptualization:** Amare Dessalew.

**Data curation:** Amare Dessalew.

**Investigation:** Shimelis Sishah.

**Methodology:** Shimelis Sishah, Getasew Azene.

**Supervision:** Shimelis Sishah, Temesgen Abrahem.

**Writing – original draft:** Temesgen Abrahem.

**Writing – review & editing:** Shimelis Sishah, Hurgesa Hundera.

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
