## [Decision Letter · Decision Letter 0]

29 Apr 2022

PONE-D-22-08462Downscaling and validating SMAP soil moisture using a Machine learning algorithm over the Awash River basin, Ethiopia.PLOS ONE

Dear Dr. sishah,

Thank you for submitting your manuscript to PLOS ONE. After careful consideration, we feel that it has merit but does not fully meet PLOS ONE’s publication criteria as it currently stands. Therefore, we invite you to submit a revised version of the manuscript that addresses the points raised during the review process. The manuscript must be corrected in all points indicated by the reviewers, such as:

- Review the entire manuscript to an expert English speaker.

- Examine soil moisture and other influencing climate factors over time.

- What criteria did the researchers follow to determine the Sample Size for this study?

- The overall quality of figures must be improved.

- State the objectives of the study clearly within abstract section.

- Clarify the rationale of Objective 1 with the hypothesis of this study. This is neither clear in introduction nor in discussion.

- Clarify soil sampling methods in detail.

- Relate the facts of results with the scientific references under discussion.

We look forward to receiving your revised manuscript.

Kind regards,

Claudionor Ribeiro da Silva

Academic Editor

PLOS ONE

Journal Requirements:

2. In your Methods section, please provide additional information regarding the permits you obtained to collect samples for the present study. Please ensure you have included the full name of the authority that approved the field site access and, if no permits were required, a brief statement explaining why.

"NO"

Reviewers' comments:

Reviewer's Responses to Questions

**Comments to the Author**

1. Is the manuscript technically sound, and do the data support the conclusions?

Reviewer #1: Yes

Reviewer #2: Partly

2. Has the statistical analysis been performed appropriately and rigorously? 

Reviewer #1: Yes

Reviewer #2: No

3. Have the authors made all data underlying the findings in their manuscript fully available?

Reviewer #1: Yes

Reviewer #2: No

4. Is the manuscript presented in an intelligible fashion and written in standard English?

Reviewer #1: Yes

Reviewer #2: No

5. Review Comments to the Author

Reviewer #1: Dear Editor,

Thank you for considering me as a reviewer for this publication in your esteemed journal of PLOS ONE. This manuscript will be ready for publication when these questions and commands are cleared.

Questions and comments:

1- Insufficient period to measure the impact of climate change on Soil Moisture Active.

2- Researchers should examine soil moisture and other influencing climate factors over time.

3- The study of estimating the roughness of the soil surface is not clear and limited!

4- The researchers did not use the Partial Least Squares regression (PLS) in this research; why?

5- What criteria did the researchers follow to determine the Sample Size for this study?

6- The overall quality of figures must be improved: labeling, font size, etc.

Reviewer #2: The present work of “Downscaling and validating SMAP soil moisture using a Machine learning algorithm over the Awash River basin, Ethiopia” the authors have presented a decent work of fine-tuning databases as per requirements through statistical procedures, thereby making this research paper purely of statistical background. However, there are ample scopes of improvements as follows-

Please add continuous line numbers to the whole text for easy detection of modifications and review the entire manuscript to an expert English speaker or a Native English speaker.

A. Abstract Section:

1. “Microwave Remote sensing”- Please mention wavelength/s/range.

2. “Coarse spatial resolution”- Mention the resolution

3. We will be grateful for an explanation of how a strong spatial correlation with the original database helps to deliver “More” soil moisture information than the original one?

4. “Cm3.Cm-3”, is this unit correct? Please explain.

5. Please avoid using abbreviations in the keyword section. Please use full forms then use abbreviations (Like MODIS in introduction section).

6. State the objectives of the study clearly within abstract section as stated at the end of the introduction section.

B. Introduction Section:

1. Please mention active, passive and active-passive satellite separately under a list.

2. Please differentiate soil moisture monitoring with hydrological monitoring... Is this paper for hydrological monitoring?

3. “However, the complex non-linear relationship between coarse-scale soil moisture product and land surface variables constrained the applicability of these downscaling approaches [13].”- Please explain this sentence properly.

4. ……..”algorithm proposed by [14] and later modified by….” This is not ver scientific way of writing references. Please mention the name, year etc properly. Rectify these types of mistakes from the other parts of the manuscript as well.

5. Please clarify the rationale of Objective 1 with the hypothesis of this study. This is neither clear in introduction nor in discussion.

C. Materials and Method:

1. Please clarify soil sampling methods in detail (depth, season, sampling machinery and lat-long of sampling sites)

2. Please mention proper references for “Satellite Data” section.

D. The authors have done quite excellent job in the result section. However, the Discussion section is very poor and doesn’t meet the objectives at all. It seems it has been written in a hurry.

Please relate the facts of results with the scientific references under discussion. Build 3 different segments for 3 different objectives and a common portion summarizing all three objectives.

Construct the conclusion accordingly as well.

6. PLOS authors have the option to publish the peer review history of their article (what does this mean?). If published, this will include your full peer review and any attached files.

Reviewer #1: No

Reviewer #2: **Yes: **PRAVAT UTPAL ACHARJEE

---

## [Author Response · Author response to Decision Letter 0]

2 Nov 2022

Awash Basin Development office provides field data collection instruments and we acknowledge the office at the end of the paper under acknowledgement section.

Regarding funding issue;

We the authors assure you to that 'The authors received no specific funding for this work.'

---

## [Decision Letter · Decision Letter 1]

19 Dec 2022

Downscaling and validating SMAP soil moisture using a Machine learning algorithm over the Awash River basin, Ethiopia.

PONE-D-22-08462R1

Dear Dr. sishah,

We’re pleased to inform you that your manuscript has been judged scientifically suitable for publication and will be formally accepted for publication once it meets all outstanding technical requirements.

Kind regards,

Claudionor Ribeiro da Silva

Academic Editor

PLOS ONE

Additional Editor Comments (optional):

Reviewers' comments:

Reviewer's Responses to Questions

**Comments to the Author**

1. If the authors have adequately addressed your comments raised in a previous round of review and you feel that this manuscript is now acceptable for publication, you may indicate that here to bypass the “Comments to the Author” section, enter your conflict of interest statement in the “Confidential to Editor” section, and submit your "Accept" recommendation.

Reviewer #2: All comments have been addressed

Reviewer #3: All comments have been addressed

2. Is the manuscript technically sound, and do the data support the conclusions?

Reviewer #2: Yes

Reviewer #3: Yes

3. Has the statistical analysis been performed appropriately and rigorously? 

Reviewer #2: Yes

Reviewer #3: Yes

4. Have the authors made all data underlying the findings in their manuscript fully available?

Reviewer #2: Yes

Reviewer #3: Yes

5. Is the manuscript presented in an intelligible fashion and written in standard English?

Reviewer #2: Yes

Reviewer #3: Yes

6. Review Comments to the Author

Reviewer #2: All comments have successfully been addressed by the authors. The manuscript is now technically sound, statistically at per and is fit to be accepted for publication as per observation from the reviewer.

Reviewer #3: Review Comments

The manuscript is well constructed and covers scientific aspects. The results of current study are interesting and will be fruitful for the readers of the Journal. However, some minor shortcomings are required to be addressed to improve the quality of the manuscript which are suggested below

1. Elaborate the objectives of the study at the end of the Introduction section.

2. Add relevant reference at the biochemical attributes in Material and Methods section

3. Include information regarding geographical location of the experimental site

4. Remove typo grammatical errors

7. PLOS authors have the option to publish the peer review history of their article (what does this mean?). If published, this will include your full peer review and any attached files.

Reviewer #2: **Yes: **PRAVAT UTPAL ACHARJEE

Reviewer #3: No

---

## [Editor Report · Acceptance letter]

22 Dec 2022

PONE-D-22-08462R1 

Downscaling and validating SMAP soil moisture using a Machine learning algorithm over the Awash River basin, Ethiopia. 

Dear Dr. Sishah:

I'm pleased to inform you that your manuscript has been deemed suitable for publication in PLOS ONE. Congratulations! Your manuscript is now with our production department. 

Kind regards, 

on behalf of

Dr. Claudionor Ribeiro da Silva 

Academic Editor

PLOS ONE